# The Anxiety Burden in Patients with Chronic Intestinal Failure on Long-Term Parenteral Nutrition and in Their Caregivers

**DOI:** 10.3390/nu16081168

**Published:** 2024-04-14

**Authors:** Lidia Santarpia, Raffaella Orefice, Lucia Alfonsi, Maurizio Marra, Franco Contaldo, Fabrizio Pasanisi

**Affiliations:** Internal Medicine and Clinical Nutrition, Department of Clinical Medicine and Surgery, Federico II University Hospital, 80131 Naples, Italy; raffyrfc7@gmail.com (R.O.); lucia.alfonsi@unina.it (L.A.); marra@unina.it (M.M.); pasanisi@unina.it (F.P.)

**Keywords:** anxiety burden, home parenteral nutrition, chronic intestinal failure, caregivers

## Abstract

**Background and aims**: Home parenteral nutrition (HPN) is a life-saving treatment for patients affected by chronic intestinal failure (CIF). Both this clinical condition and its therapy require radical lifestyle modifications, affecting life quality and psychological balance in patients as well as family members. Patient psychological burden has rarely been taken into consideration, not to mention that of caregivers. This study aims to evaluate the levels of anxiety in CIF patients on HPN, and their caregivers, consequently determining their impact on the psychological and physical aspects. **Methods:** After a brief introductory interview, adult patients on HPN for CIF and their caregivers were asked to fill in the HAMA-A questionnaire. **Results:** Fifty patients and their respective caregivers were enrolled. Mean HAMA-A scores were similar in patients and caregivers and testified the presence of a mild to severe impact of CIF and HPN in both groups, with a significantly higher impact on female patients and caregivers. After adjusting age, education level, duration of CIF and HPN dependence, and degree of kinship, no differences were revealed in the scores. **Conclusions:** The study confirms that CIF patients on HPN and their caregivers have a significant anxiety burden independently from the duration of the disease, therefore needing appropriate support.

## 1. Introduction

Chronic intestinal failure (CIF) is a severe malabsorptive condition in which patients become dependent on home parenteral nutrition (HPN) to assure their nutritional and fluid requirements [1,2].

In general, this condition occurs almost unexpectedly due to intestinal ischemia, Crohn’s disease, and other pathologies, significantly affecting the lives of patients and family members. Indeed, patients immediately find themselves in a new anatomical and clinical condition attributable to the loss of their bowel function. Patients initially feel disbelief, anxiety, hopelessness, and grief ascribable to the loss of freely eating and drinking, together with problems in body image, work, and family relationships [3,4].

Patients with CIF must struggle against a life-threatening disease and may be worried about repeated surgeries and hospital admissions, experiencing fatigue which interferes with their daily life activities [5]. Fatigue is a complex concept that may include physical, social, and psychological dimensions [6], and may also be associated with anxiety and depression [5]. Finally, patients may have a reduced working capacity since they have to manage HPN therapy, eating and drinking medical prescriptions, intestinal functions, and eventually a stoma [5,6].

HPN dependence influences patient autonomy, work ability, and capacity to travel and socialize [7]; moreover, sleep quality is significantly compromised due to the administration of therapy during night hours [8,9,10]. Most forms of CIF are often lifelong conditions with HPN dependency for many years. On the other hand, HPN infusions can last up to 12 h and longer in some instances, therefore affecting everyday life.

In addition, although life sustaining for CIF patients, HPN may be hampered by serious complications, often requiring hospitalization [9], including catheter-related bloodstream infections, thrombosis, embolism [9,10,11], and long-term hepatic failure and osteoporosis [12].

Affected by such a severe disease, patients almost always need the help of their family members [5]. For this reason, it is important to consider the psychological, economic, and social strains on caregivers who have to re-organize their life and work according to the new family condition. Initially, they may feel a certain amount of responsibility, stress, and anxiety in learning how to manage HPN and eventual consequential complications [13].

However, because of the low CIF prevalence, the burden that caregivers face has not been adequately characterized.

The objective of this study was to investigate the burden experienced by adult caregivers of adult patients with CIF. Caregivers may experience fatigue, sleep disturbance, and some strain and stress, with markedly reduced productivity; furthermore, caregiving responsibilities may limit time with family, friends, and recreational activities.

The levels of anxiety and stress in patients and their caregivers may influence the approach to the disease and to HPN, negatively affecting life quality [11,12,14,15].

To the best of our knowledge, very little emphasis has been placed on the psychological issues of CIF patients on HPN and even less on their caregivers.

## 2. Aims of the Study

The present study aims to measure the anxiety burden in CIF patients on HPN and in their caregivers. A secondary aim is to search for possible correlations between anxiety levels and disease duration, education, employment, and degree of kinship between patients and caregivers.

## 3. Patients and Methods

### 3.1. Study Population

All consecutive CIF patients on HPN and their caregivers, meeting the following inclusion and exclusion criteria, were asked to participate in this study.

### 3.2. Inclusion Criteria

-CIF patients and caregivers aged between 18 and 70;-CIF patients on HPN from at least 1 month;-Signed informed consent.

### 3.3. Exclusion Criteria

-Drug addiction or abuse;-Psychosis or personality disorders classified according to the DSM-V criteria;-Acute medical complications requiring hospitalization.

After having signed the informed consent, during a brief interview, data on age, employment, education, and degree of kinship were collected both for patients and their caregivers to evaluate a possible correlation with anxiety levels.

As far as education, patients and caregivers were classified as having completed primary school, secondary school, high school, or a university degree. As far as kinship, people were categorized as follows: husband, wife, son/daughter, mother/father, and brother/sister. Employment was identified as remunerated or non-remunerated, and the latter comprised housewives and unemployed people; in addition, the category of retirement was also identified.

Information on disease type and duration and HPN dependence was obtained from the patients’ medical records.

Patients and caregivers took part in an introductory meeting held with a dedicated psychologist to receive instructions on the modality to fill out the Hamilton Anxiety Rating Scale (HAM-A) questionnaire.

All subjects signed an informed consent for inclusion before participating in the study. The study was conducted in accordance with the declaration of Helsinki and approved by the Ethics Committee of A.O.U. Federico II—AORN Cardarelli (ethical approval date: 24 October 2018; prot. n: 317/18).

### 3.4. The Hamilton Anxiety Rating Scale (HAM-A)

The Hamilton Anxiety Rating Scale (HAM-A) [16], created by Hamilton in 1959, is still today one of the most widely used rating scales to measure the severity of perceived anxiety symptoms [17,18,19]. It is a clinician-rated evaluation, consisting of 14 symptom-defined elements, for both psychological and somatic symptoms.

Each item, based on a five-point ratio scale, is scored independently according to the intensity of symptoms perceived by the patients: 0 indicates an absence of feeling; 1 is equivalent to mild, 2 to moderate, 3 to severe, and 4 to very severe symptoms.

Once the evaluation is completed, the clinician sums up the scores based on the 14 individually rated items, yielding a result ranging from 0 to 56.

HAM-A cut offs are: ≥18—mild anxiety; ≥19–25—moderate anxiety; ≥19–25—moderate anxiety.

The 14 single items are as follows:**Anxious mood** (worries, fear of the worst, irritability);**Tension** (startle response, fatigability, restlessness, easy tearing, trembling, inability to relax);**Fear** (of the dark/traffic/crowds, of being left alone);**Insomnia** (difficulty in falling asleep, broken sleep, unsatisfying sleep and fatigue on waking, dreams, nightmares, night terrors);**Cognitive functions** (poor memory and difficulty in concentration);**Depressed mood** (loss of interest, lack of pleasure in hobbies, depression, shifting moods);**Somatic muscular symptoms** (aches and pains, stiffness, bruxism, twitching, myoclonic jerks, teeth grinding, unsteady voice, increased muscular tone);**Sensory** (tinnitus, blurred vision, hot and cold flushes, feelings of weakness, pricking sensation);**Cardiovascular** (tachycardia, extrasystoles, palpitations, chest pain, vessel throbbing, faint feeling);**Respiratory** (chest tightness, choking pressure, sighing, dyspnoea);**Gastrointestinal** (irritable bowel syndrome-type symptoms, abdominal colicky pain, burning sensations, abdominal fullness, swallowing difficulty, nausea, vomiting, borborygmi, looseness of bowels/constipation, weight loss);**Genitourinary** (urinary frequency or urgency, amenorrhea, menorrhagia, frigidity, premature ejaculation, loss of libido, impotence);**Autonomic** (dry mouth, tension headache, flushing, sweating, giddiness, goose bumps);**Observed behaviour at interview** (fidgeting, restlessness or pacing, tremor of hands, furrowed brow, strained face, sighing or rapid respiration, pallor, swallowing, etc.).

### 3.5. Statistical Analysis

Data were included in a database, expressed as mean ± SD, when appropriate, and analysed with SPSS ver. 28. The *t*-test was used for the comparison of continuous variables, while the Mann–Whitney test was used for the comparison of parametric variables. The significance level was reached with *p* values of *p* < 0.05.

## 4. Results

In total, 50 patients (28 F; 22 M) and 50 respective caregivers (34 F; 16 M), both aged between 18 and 70, were enrolled. Personal (age, sex) and social (work, education) information of both are reported in Table 1.

As far as residual small bowel length, 7 patients did not undergo resections, while 43 patients had a median remnant small bowel of 108 cm (range of 15–300). In 38 patients, the small bowel was in continuity with the colon and, consequently, regularly canalized (no stoma). The percentage of colon in continuity was calculated according to the Cummings percentage [20]. Seven patients had an ileocecal valve, and twelve patients had a stoma.

The total HAM-A score revealed the presence of anxiety both in patients and caregivers, with a similar distribution among the mild, moderate, and severe levels in the two groups. None of the differences between groups were significant (Figure 1).

Moreover, when dividing the groups according to gender, it can be seen that female patients had a significantly lower prevalence of mild (*p* = 0.001, Mann–Whitney test) and a higher prevalence of moderate (*p* = 0.004) anxiety levels than males (Figure 1).

In a similar way, although not significantly, in female caregivers, there was a trend for a higher prevalence of moderate and severe anxiety levels than males (Figure 1).

Education level and employment type did not influence the degree of anxiety.

The analysis of the single items of the HAM-A scale in patients and caregivers revealed the following (Figure 2):

Anxious mood: even if not statistically significant, there was a trend for higher rates of absent and mild anxiety in patients than caregivers.

The distribution of the tension levels was significantly different in the two groups; indeed, tension was absent in 14% of patients, while severe and very severe levels were mainly represented in the caregiver group.

Fear is almost absent (58 vs. 32%) or mild (22% vs. 8%) in patients, and moderate (24 vs. 8%) or severe (24 vs. 4%) in caregivers, with a significant different distribution among groups (*p* = 0.009).

As far as insomnia, there was no significant difference between the distribution levels in patients and caregivers, except for a higher trend of mild forms (22 vs. 4%) in patients and moderate forms in caregivers (34 vs. 12%).

Similarly, both intellectual functions and depressed mood are equally distributed in patients and caregivers, without significant differences.

Regarding somatic symptoms (muscular tension), 50% of caregivers (vs. 28% of patients) had no symptoms, and 16% of patients (vs. 6% of caregivers) had very severe symptoms, with a significantly different distribution (*p* = 0.026) among the two groups.

Also, sensory alterations were differently distributed between the two groups (*p* = 0.042) with 52% of caregivers (vs. 32% of patients) having no symptoms.

Cardiovascular and respiratory symptoms were almost absent or mild in both patients and caregivers, with a similar distribution in the two groups.

Gastrointestinal symptoms were absent in 58% of caregivers (vs. 16% patients), while they were present in most patients, with a different intensity. The calculated differences were significant (*p* = 0.003).

Similarly, genitourinary symptoms were absent in 78% of caregivers (vs. 52% of patients), while patients had diverse manifestation intensity with a meaningful different distribution between the two groups (*p* = 0.021).

Autonomic symptoms have analogous distribution in patients and controls; in 50% of both groups, an absence of symptoms was observed, whilst severe and very severe symptoms were found in a minority of patients (8%) and caregivers (12%).

Regarding the last item, observed behaviour at interview was absent in 14% of patients and 4% of caregivers, while for the other categories, the distribution was not significantly different in both groups.

## 5. Discussion

From the questionnaires completed, a mean moderate anxiety for disease and therapy was found in most patients and caregivers, with an approximate homogeneous distribution on the three levels. CIF patients are a group with heterogeneous clinical conditions, for many factors, including intestinal length, the presence of a stoma, the average volume of the parenteral nutritional support administered, and the frequency per week. The psychological state of patients was mostly derived from the awareness of their disease, the new anatomical and clinical conditions, HPN dependence, and worries regarding their future [1,8,13]. On the other hand, the new patient needs influenced the daily and working lives of caregivers [1,8,13]. Caregivers do not choose to care, but this happens because they have an emotional connection with the person requesting assistance.

Anxiety occurred independently from the type and duration of the disease, relation of kinship, education, and employment, confirming that both CIF and HPN represent an unexpected destabilization that chronically impacts daily life and lifestyle, and to which both patients and caregivers adapt with difficulty. Indeed, by dividing patients according to disease duration (< or >5 y), no significant differences emerged between groups, testifying that no adaptation to this condition occurred over time.

As far as gender, women showed higher emotional involvement than men in facing difficulties, both as caregivers and patients.

In the Italian culture, women have a primary role in caring for the family, and they are even further loaded in their care giving function when a member is affected by an illness. Often there is a tendency to lose the original relational function with the family member, i.e., a wife caregiver tends to change from a wife to a mother/child kinship, causing her discomfort and anxiety. Differently, a male caregiver generally tends to preserve his role [21,22,23].

Compared with males, female patients give a higher value and significance to their body image and function, which deeply influences their psychological sphere [24].

Our data are in contrast with the survey by Castinel J et al. [25] conducted in a French nutrition referral centre on patients with CIF, treated with HPN, and their caregivers; the authors found that male caregivers experienced a heavier burden than women. We are not able to explain the reasons for these differences in the two studies, but they are probably due to the different importance of the woman’s role in the family, as a consequence of cultural traditions.

Nonetheless, our findings agree with studies in the literature and highlight the importance of identifying and addressing the needs of patients and caregivers.

A retrospective multicentre cohort study by Silva R et al. [26] assessed the burden of CIF in Portuguese patients on parenteral nutrition (PN). Twenty of the thirty-one evaluated patients were paediatric. The author found that patient management was characterized by a substantial therapeutic burden and healthcare resource use, translating into high direct costs and a substantial impairment of the adult physical functions and the child school functions.

The article by Smith CE et al. [27] collected opinions from teens and young adults with CIF requiring long-term HPN on the following issues: illness-related stressors, daily management of HPN, and its effects on their everyday lives. These young patients struggled to live with gastrointestinal symptoms and multiple disease exacerbations requiring hospitalizations. When compared to those with similar medical conditions but without HPN, teen and young adult patients report psychological and social problems. Rather than accepting this lifelong procedure, young patients, desiring alternative restoring solutions, search for answers to their questions on various levels.

Finally, this type of study should be extended to other groups of patients with chronic diseases needing the help of a family caregiver. For example, Kurita GP et al. [28] evaluated three Danish populations of caregivers of patients with different life-threatening pathologies (renal failure, cystic fibrosis, and CIF). All three caregiver groups had a self-perception of poor energy and health, investing many hours into the care role.

The evaluation performed in the present study is part of a future aim consisting of the organization of focused therapeutical support carried out by a psychologist providing the necessary instruments for anxiety management for both patients and caregivers.

In the following ongoing phase, the systemic-relational method will be used, focusing on the preservation of the relational function between patient and family member caregiver, which generally is modified due to the disease [29].

Systemic-relational psychotherapy considers the subject within the family. The family is a unit where each member has a specific role. According to this theory, critical events like a separation, a severe disease, or grief do not represent an individual issue, but one that influences the whole family system. In the presence of a disease, deep psychological and structural transformations happen and may generate not only a crisis to the individual, but a complete unbalance of the whole family.

In addition to the global result, each of the 14 items was evaluated, revealing that caregivers appeared, for some aspects, more psychologically involved. The data obtained from the first three items (anxiety, tension, and fear) led us to hypothesize that patients were inclined to transfer these burdens onto their caregivers. The next three items—insomnia, altered intellectual functions, and depressed mood—were consistently present and equally distributed in both groups. Insomnia might partially derive from the nocturnal infusion of the nutritional mixture; moreover, sleeplessness significantly influences daily attention, concentration, and performance.

Furthermore, the somatic symptom items were more emphasized in patients, going from muscular tension to sensory alterations, more specifically gastrointestinal and genitourinary symptoms. Gastrointestinal disturbances were most probably influenced by intestinal insufficiency, responsible for urgent evacuations, intestinal rumours, and abdominal pain. Genitourinary complaints were influenced by nocturnal HPN, responsible for increased diuresis.

The items regarding cardiovascular and respiratory symptoms showed no differences between patients and caregivers.

## 6. Study Limitations

This is a monocentric study, with a relatively small number of participants due to the rarity of the disease. For this reason, findings often were not statistically significant and might not be generalizable to other settings, also because the management of this rare disease is non-homogeneous in the different Italian and European centres.

Moreover, we did not use a specific test for CIF patients on HPN [8]. Indeed, HAM-A is a non-selective/specific test, extendible to different study populations, and it was used in the present work as part of a broader investigation that included different types of patients.

As already stated, the comparison of gastrointestinal symptoms between patients and caregivers may be a limitation of the HAM-A questionnaire because patients with CIF frequently have diarrhoea, abdominal pain, etc.; nevertheless, this is one of the fourteen points in the questionnaire.

Finally, a control group (patients with chronic gastrointestinal disease but not on HPN) would have been interesting, but unfortunately, our outpatient unit is specifically for adult patients with CIF on HPN.

In the literature, only a few studies are available on the quality of life of CIF patients on HPN and their family members; additionally, the tools used were multiple and differentiated according to the aims (emotional, social, or economic burden, etc.). Furthermore, most studies are focused on oncological or paediatric patients [30,31,32,33,34].

## 7. Conclusions

Despite these limitations, this is one of the few studies evaluating the impact of CIF and HPN not only on patients, but particularly on family members providing care, evidencing a significant influence on the quality of life of both and the importance of psychological counselling and care.

We believe that these results may be of inspiration for a multicentre study extendible to other European CIF Centres.

As already mentioned, it would be of extreme help and interest to evaluate the anxiety burden of caregivers, extending this new future evaluation to other fields of chronic and life-threatening illnesses, particularly those involving lengthy procedures.

## Figures and Tables

**Figure 1 nutrients-16-01168-f001:**
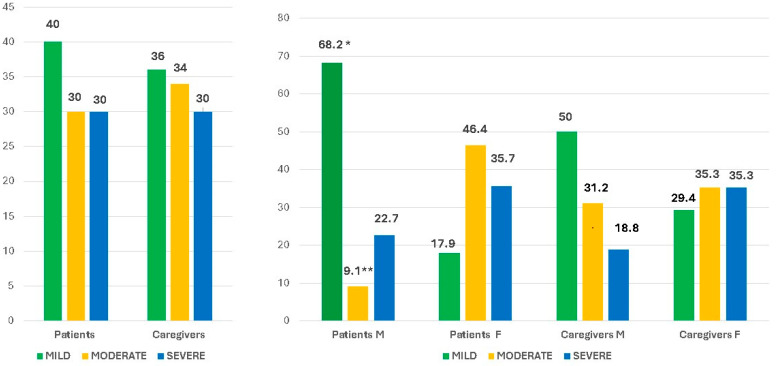
Total HAM-A score in patients and caregivers both in the whole group and according to gender. * *p*= 0.001 M vs F; ** *p* = 0.004 M vs F.

**Figure 2 nutrients-16-01168-f002:**
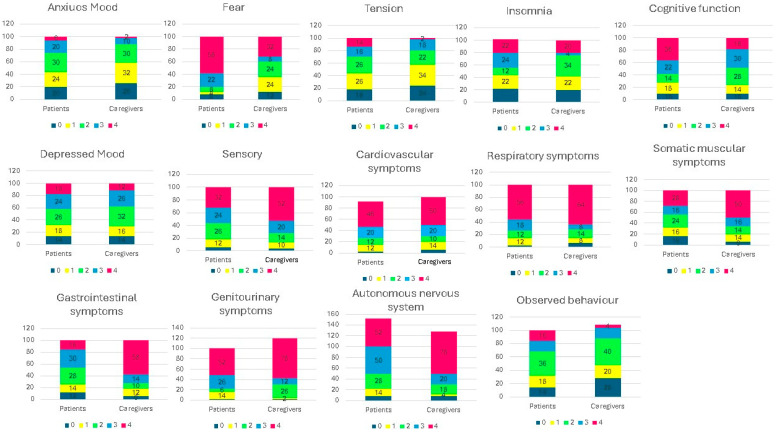
Clinician-rated evaluation of the 14 items: anxious mood, tension, fear, insomnia, cognitive functions, depressed mood, somatic muscular symptoms, sensory, cardiovascular, respiratory, gastrointestinal, genitourinary, autonomic, observed behaviour at interview. Each item is based on a five-point ratio scale and is scored independently according to the intensity of what the patients felt: **0** = total absence of symptoms; **1** = mild symptoms; **2** = moderate symptoms; **3** = severe symptoms; **4** = very severe symptoms. Tension *p*: 0.027; fear: 0.009; Mann–Whitney test; somatic muscular symptoms: *p* = 0.024; sensory: *p* = 0.042; gastrointestinal *p* = 0.001; genitourinary *p* = 0.021.

**Table 1 nutrients-16-01168-t001:** Personal and social information of patients and caregivers.

	Patients (n. 50)	Caregivers (n. 50)	*p*
Age (years)	48.4 ± 15.0; median 52 (18–69)	50.7 ± 13.3; median 54 (21–71)	n.s
Gender	28 F/22 M	34 F/16 M	n.s
Disease duration (yrs)	6.5 ± 5.38; median 7 (0.5–22)	-	-
**Primary disease**			
Intestinal infarction	17		
Crohn’s disease	15		
Intestinal volvulus	5		
Intestinal adhesions	4		
Intestinal pseudobstruction	3		
Bariatric surgery	2		
Mucosal disease	2		
Radiation enteritis	2		
Residual small bowel length, cm median (range)	7 no resections in 43: median 108 cm (15–300)		
Presence of stoma (yes/no)	12/38		
Presence of ileocecal valve (yes/no)	7/43		
Colon in continuity (yes/no)	38/12		
Percentage of colon in continuity according to Cummings’ classification median (range)	70 (0–100)		
PN dependency duration (yrs)	6.5 ± 5.38; median 7 (0.5–22)		
Days of infusion per week	5 (3–7)		
Weight (kg)	55.9 ± 10.5 (35–80)	-	-
BMI (kg/m^2^)	20.7 ± 3.2 (12.5–25.5)	-	-
**Degree of kinship**			
Husband	16	14	
Wife	15	16	
Son/daughter	11	6	
Mother/father	6	11	
Brother/sister	3	3	
**Education**			
Primary school license	5	3	
Secondary school license	17	18	
High school license	23	23	
University degree	5	4	
Employment	19 remunerated	23 remunerated	
	26 non remunerated	23 non remunerated	
	5 retired	4 retired	
Total HAM-A score	20.94 ± 10.72 (3–42)	21.94 ± 10.85 (7–52)	n.s

## Data Availability

Data available on request due to privacy and legal/ethical reasons.

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
