# Peer review of "The Anxiety Burden in Patients with Chronic Intestinal Failure on Long-Term Parenteral Nutrition and in Their Caregivers"

_nutrients, 2024, doi:10.3390/nu16081168_

Round 1
Reviewer 1 Report
Comments and Suggestions for Authors
Author Response
Reply to reviewer 1:
The authors thank the reviewer for the useful comments allowing improvement of the manuscript.
This study by Orefice R and colleagues describes the anxiety levels of patients and caregivers with Chronic Intestinal Failure. The strenght of the study is scarce literatura in this field. Nevertheless, there are some considerations the authors need to review.
MAJOR COMMENTS:
Methods:
Explain in this section the variables recorded in the results (i.e., education, etc). Explain what means non remunerated employment.
As suggested, the variables recorded have been explained in the text.
The meaning of “non remunerated employment” (i.e: homemaker, student) has been explained. Moreover, the category of retired has been added.
Results:
Consider to calculate the median of the age of patients and the duration of the disease as there is a wide range of the time in these two variables.
The median age of patients (52 years) and caregivers (54 years) and the duration of the disease (7 years) have been calculated and added in table 1.
Table 2: consider to delete the abreviation pts in some boxes of the table (residual small bowel length) as there is no this abreviation in another box (primary disease).
As suggested, the abbreviation “pts” has been deleted
Explain more in detail in this section or in methods what exactly means colon in continuity (is different that presence of stoma?).
We have better detailed in the results the meaning of colon in continuity, explaining that patients with colon in continuity do not have a stoma.
Data regarding days of infusion/week is no clear: means days or weeks?
Data regarding days of infusion means days per week, now clarified in the table.
Consider add information regarding the presence or no of ileocecal valve.
Information regarding the presence or not of ileocecal valve has now been added in the table.
Table 3: consider to explain this information with a graphic instead of a table. Also Tables 4a and 4b could be explained as a graphic (column chart or bar diagram).
As suggested, the information in tables 3 and 4 (a,b) are now represented as a graphic (new figure1).
Some of the gastrointestinal symptoms presented in the HAM-A scale could be secondary to the disease, please consider this situation in the results (explain the gastrointestinal symptoms in relation of primary disease) and in the discussion section.
As evidenced in the discussion section, some of the gastrointestinal symptoms presented in the HAM-A scale could be secondary to the disease.
Consider to add a table or the explanation about secondary analysis: difference in HAM-A score in relation to gender, time of the diseas, primary disease. It would be usefull to add this information with respect of the diferent variables studied.
Unfortunately, the brief report “regulation” does not allow more than 1 table; however, this detailed information can be found in the result section.
Discussion:
It would be usefull to explain this section with an homogeneus structure. An optional structure could be with these paragraphs:
Explain briefly the results of the study and compare with literature.
Explain the diference in caregivers vs patients. o Information regarding gender.
Explain the future aim of this study.
Differences in somatic symptoms (specifically gastrointestinal symptoms) between patients and caregivers should be interpretated as a limitation of HAM-A questionnaire in patiens with chronic intestinal failure.
Another limitation of the study is the lack of control group. That means that patients with these chronic gastrointestinal disease buy without intestinal failure (i.e: not depend on chronic parenteral nutrition) could have similar or different HAM-A results.
The discussion has been re-organized according to the suggested sequence.
We clarified that the comparison of gastrointestinal symptoms between patients and caregivers may be a limitation of the HAM-A questionnaire because patients with chronic intestinal failure frequently have diarrhoea, abdominal pain, etc. This subject is one of the 14 points in the questionnaire.
Moreover, we evidenced that a control group would have been interesting, but our outpatient unit is specifically for patients with intestinal failure on HPN. This limitation has been added in the study limitation section.
Consider add a conclusion sentence at the end of the discussion section.
We considered adding a conclusion sentence which can be found at the end of the discussion.
Female gender is not associated with worse results in anothers studies. This could be a useful reference to explain:
Male gender is associated with informal caregiver burden in patients with chronic intestinal failure treated with home parenteral nutrition. Castinel J, Pellet G, Laharie D, Zerbib F, Silvain C, Wilsius E, Kerlogot L, Rivière P, Poullenot F.JPEN J Parenter Enteral Nutr. 2022 Sep;46(7):1593-1601. doi: 10.1002/jpen.2340. Epub 2022 Feb 23.PMID: 35092023
Our data are in contrast with the survey by Castinel J et al., made in a French nutrition referral centre on patients with CIF treated with HPN and their caregivers; the authors found that male caregivers experienced a heavier burden than women. We are not able to explain the reason of these differences, probably due to different family culture. In Italy, women, particularly those from the south are extremely apprehensive and protective towards their family.
MINOR COMMENTS:
Introduction:
there is a ¨,¨ in red color between ¨patient autonomy¨ and ¨work ability¨.
on CIF patients on HPN and even les son their caregivers : is in another colour.
Methods:
In Statistical analysis seems that there is a double space between ¨variables¨ and ¨.The significance…¨
Results:
Page 5, below the table: there are some words in italics and the word mild is underlined.
Words in italics and the underlined word “mild” have been corrected.
Page 6: insignificant→ consider to write ¨not statistically significant¨.
Discussion: • Page 6: move the dot between ¨caregivers¨ and (1,6, 12).
Page 6: compared with male is wrote in italics
Reviewer 2 Report
Comments and Suggestions for Authors
Very nicely written manuscript addressing the important topic of anxiety among patients with chronic intestinal failure patients and home parenteral nutrition dependency as well as their caregivers. I believe if you conduct an extensive literature review you will find decades of papers addressing psychosocial support in this population, as far back as 1979 and in 1980's; although the field did not use the term CIF and many of these paper address home parenteral nutrition, albeit, HPN dependent patients typically had intestinal failure. Nevertheless, the papers cited are current and relevant. Caregiving has gained new interest and there are many more current reports. Here are a few that the authors may wish to review.
Castinel J et al. JPEN 2022. 46(7):1593-1601. Male gender is associated with informal caregiver burden in patients with chronic intestinal failure treated with home parenteral nutrition.
Silva R et al. Port J Gastroenterol 2022. 30(4):293-304. Clinical, economic, and humanistic impact of short-bowel syndrome/chronic intestinal failure in Portugal (Parenteral Study).
Kurita GP et al. Palliat Support Care 2023. 27:1-7. The impact of caring on caregivers of patients with life-threatening organ failure [the organ failure is CIF]
Smith CE et al. JPEN 2021. 45(3):499-506. Themes of stressors, emotional fatigue, and communication challenges found in mobile care discussion sessions with patients requiring lifelong home parenteral nutrition infusions.
I have a few minor comments for the authors to address.
1) Please report length of PN dependency. The inclusion criteria suggests patients with >1 month of PN were eligible to participate. The discussion mentions that there was no difference in anxiety levels by disease duration < or > 5 years, but data are not reported.
2) Did any of the patients or caregivers have anxiety before the onset of disease or need for PN? Any other healthcare conditions as co-morbidities that may relate to anxiety?
3) Do you have data for how many patients provide independent self care for HPN related procedures, or do they all rely on family caregiver for this?
4) Table 1 - please move the descriptors for disease/diagnosis, degree of kinship, education level, employment to Column 1 and only report numerical value (n, %) in column 2 and column 3.
Comments on the Quality of English LanguageMinor copy editing required.
Author Response
The authors thank the reviewer for the useful comments allowing improvement of the manuscript.
Very nicely written manuscript addressing the important topic of anxiety among patients with chronic intestinal failure patients and home parenteral nutrition dependency as well as their caregivers. I believe if you conduct an extensive literature review you will find decades of papers addressing psychosocial support in this population, as far back as 1979 and in 1980's; although the field did not use the term CIF and many of these paper address home parenteral nutrition, albeit, HPN dependent patients typically had intestinal failure. Nevertheless, the papers cited are current and relevant. Caregiving has gained new interest and there are many more current reports. Here are a few that the authors may wish to review.
Castinel J et al. JPEN 2022. 46(7):1593-1601. Male gender is associated with informal caregiver burden in patients with chronic intestinal failure treated with home parenteral nutrition.
Silva R et al. Port J Gastroenterol 2022. 30(4):293-304. Clinical, economic, and humanistic impact of short-bowel syndrome/chronic intestinal failure in Portugal (Parenteral Study).
Kurita GP et al. Palliat Support Care 2023. 27:1-7. The impact of caring on caregivers of patients with life-threatening organ failure [the organ failure is CIF]
Smith CE et al. JPEN 2021. 45(3):499-506. Themes of stressors, emotional fatigue, and communication challenges found in mobile care discussion sessions with patients requiring lifelong home parenteral nutrition infusions.
Thank you for the pleasant comments on our study and for the interesting suggestions: We read the suggested papers and commented them in the discussion section.
I have a few minor comments for the authors to address.
- Please report length of PN dependency. The inclusion criteria suggests patients with >1 month of PN were eligible to participate. The discussion mentions that there was no difference in anxiety levels by disease duration < or > 5 years, but data are not reported.
Length of PN dependency has been added in the text
- Did any of the patients or caregivers have anxiety before the onset of disease or need for PN? Any other healthcare conditions as co-morbidities that may relate to anxiety?
We registered among our patients and caregivers no pathological anxiety before the onset of the disease
- Do you have data for how many patients provide independent self care for HPN related procedures, or do they all rely on family caregiver for this?
Initially all patients need the help of family caregiver. Subsequently, some patients became partially independent. In our patient sample, only 5 out of 50 were able and willing to auto-provide care; at any rate, they lived with one or more supportive family members.
Table 1 - please move the descriptors for disease/diagnosis, degree of kinship, education level, employment to Column 1 and only report numerical value (n, %) in column 2 and column 3.
We moved the descriptors for disease/diagnosis, degree of kinship, education level, employment to Column 1 and the numerical values (n, %) to columns 2 and 3.
Round 2
Reviewer 1 Report
Comments and Suggestions for Authors
Manuscript ID: Nutrients-2916317.
This study by Orefice R and colleagues describes the anxiety levels of patients and caregivers with Chronic Intestinal Failure.
The strenght of the study is scarce literature in this field.
The article has been revised according to the recommendations.
However, there are some aspects that must be considered before acceptance:
· The position/order of the authors of the article has changed.
· Regarding the explanation about colon in continuity (¨patients with colon in continuity do not have a stoma¨). There are 50 patients: 38 of them do not have a stoma, but nevertheless there are 70 patients in continuous colon box. Please specify the difference.
· ¨ In Italy, women, particularly those from the south are extremely apprehensive and protective towards their family¨ à This is a subjective value judgment that perhaps should be eliminated or modified in the discussion.
· The conclusions refer to the strengths of the study, but do not summarize them (for example, they do not talk about the quality of life results found).
Author Response
The authors thank the reviewer for the additional comments made to the manuscript.
This study by Orefice R and colleagues describes the anxiety levels of patients and caregivers with Chronic Intestinal Failure.
The strength of the study is scarce literature in this field.
The article has been revised according to the recommendations.
However, there are some aspects that must be considered before acceptance:
The position/order of the authors of the article has changed
In agreement with the other authors, the order of author ordering has been changed and the request form filled out explaining the reasons.
Regarding the explanation about colon in continuity (¨patients with colon in continuity do not have a stoma¨). There are 50 patients: 38 of them do not have a stoma, but nevertheless there are 70 patients in continuous colon box. Please specify the difference.
For major clarity, a new line has been added to table 1, to detail the number of patients with colon in continuity. In this way, the percentage of colon in continuity expressed according to Cummings’ classification (ref 20) results clearer.
In Italy, women, particularly those from the south are extremely apprehensive and protective towards their family¨ à This is a subjective value judgment that perhaps should be eliminated or modified in the discussion.
As suggested, the sentence has been removed.
The conclusions refer to the strengths of the study, but do not summarize them (for example, they do not talk about the quality of life results found).
The conclusion has been modified according to the suggestion, the quality results found are now specified